

# Subterranean movement inferred by temporary emigration in Barton Springs salamanders (*Eurycea sosorum*)

Nathan F. Bendik[1], Dee Ann Chamberlain[1], Thomas J. Devitt[1,2],
Sarah E. Donelson[1], Bradley Nissen[1,3], Jacob D. Owen[1,4], Donelle Robinson[1,5],
Blake N. Sissel[1,6] and Kenneth Sparks[1,7]

[1] Watershed Protection Department, City of Austin, Austin, TX, United States of America
[2] Current affiliation: University of Texas, Department of Integrative Biology and Biodiversity Center, Austin, TX, United States of America
[3] Current affiliation: Tennessee State University, Department of Agricultural and Environmental Sciences, Nashville, TN, United States of America
[4] Current affiliation: Randolph Air Force Base, AFCEC, JBSA ISS Natural Resource Support, San Antonio, TX, United States of America
[5] Current affiliation: United States Fish and Wildlife Service, Austin Ecological Services Field Office, Austin, TX, United States of America
[6] Current affiliation: Travis County, Department of Transportation and Natural Resources, Austin, TX, United States of America
[7] Current affiliation: Baer Engineering & Environmental Consulting, Inc., Austin, TX, United States of America

Corresponding author
Nathan F. Bendik,
nathan.bendik@austintexas.gov

## ABSTRACT

Movement behavior is an important aspect of animal ecology but is challenging to study in species that are unobservable for some portion of their lives, such as those inhabiting subterranean environments. Using four years of robust-design capture-recapture data, we examined the probability of movement into subterranean habitat by a population of endangered Barton Springs salamanders (*Eurycea sosorum*), a species that inhabits both surface and subterranean groundwater habitats. We tested the effects of environmental variables and body size on survival and temporary emigration, using the latter as a measure of subterranean habitat use. Based on 2,046 observations of 1,578 individuals, we found that temporary emigration was higher for larger salamanders, 79% of which temporarily emigrated into subterranean habitat between primary sampling intervals, on average. Body size was a better predictor of temporary emigration and survival compared to environmental covariates, although coefficients from lower ranked models suggested turbidity and dissolved oxygen may influence salamander movement between the surface and subsurface. Surface population dynamics are partly driven by movement below ground and therefore surface abundance estimates represent a fraction of the superpopulation. As such, while surface habitat management remains an important conservation strategy for this species, periodic declines in apparent surface abundance do not necessarily indicate declines of the superpopulation associated with the spring habitat.

## INTRODUCTION

Amphibians are declining in many parts of the world owing to numerous factors (*Lannoo, 2005*; *Catenazzi, 2015*; *Grant et al., 2016*). Studies linking movement patterns to underlying behavioral and ecological mechanisms (*Nathan et al., 2008*) are needed to better understand amphibian population declines (*Pittman, Osbourn & Semlitsch, 2014*). Tracking individual movement and fate has been suggested as a way to close this information gap by directly linking fitness consequences of movement to management strategies or environmental conditions (*Bailey & Muths, 2019*). Although appealing, this approach requires radio telemetry (*Bailey & Muths, 2019*), which is impractical for many amphibians because they are too small to be fitted with transmitters. Indirect assessments of movement behavior are often more practical for many amphibians (e.g., capture-recapture studies; *Marsh et al., 2005*; *Lowe, Likens & Cosentino, 2006*; *Ringler, Ursprung & Hödl, 2009*; *Cayuela et al., 2020*), including fossorial or cave-dwelling species (*Balázs, Lewarne & Herczeg, 2020*).

Many salamander species use subterranean habitat because it provides environmental conditions that are relatively stable compared to surface habitats. Some species are obligate residents of caves with highly specialized adaptations (troglobites), while others are facultative troglophiles that periodically venture underground (*Gorički et al., 2019*). Studies of these populations are challenging because individuals are difficult or impossible to observe directly. Because of access restrictions to cave habitats, high levels of endemicity (*Culver et al., 2000*; *Trontelj et al., 2009*), and apparent rarity (*Krejca & Weckerly, 2007*; *Jugovic et al., 2015*; *DiStefano et al., 2020*), answering basic ecological questions (e.g., Who lives there? How many are there?) remains a fundamental challenge for the study of cave-dwelling animals (*Wynne et al., 2019*). Cave-dwelling vertebrates are typically observed only in relatively small portions of their habitat, where humans can enter or remotely access their environment. Sampling generally occurs in human-accessible caves or through "windows" into their environment such as wells or spring outlets (*Miller & Niemiller, 2008*; *Graening et al., 2010*; *Day, Gerken & Adams, 2016*; *Niemiller et al., 2016*; *Phillips et al., 2017*; *Krejca & Reddell, 2019*). Moreover, many cave-adapted species are of conservation concern because of their high endemicity and the sensitivity of subterranean habitats to disturbance and pollution (*Chippindale & Price, 2005*; *Miller & Niemiller, 2008*; *Fenolio et al., 2013*; *Devitt et al., 2019*; *Gorički et al., 2019*), underscoring the need for basic research on the biology of subterranean fauna (*Mammola et al., 2019*).

Studies of cave organisms have relied mostly on capture-recapture data to make inferences about their movement behavior (*Means & Johnson, 1995*; *Trajano, 1997*; *Lopes Ferreira et al., 2005*; *Băncilă et al., 2018*; *Balázs, Lewarne & Herczeg, 2020*). Direct observations of recaptured individuals have been used to quantify movement patterns where habitat is accessible (*Means & Johnson, 1995*; *Trajano, 1997*; *Balázs, Lewarne & Herczeg, 2020*). Where animals retreat to inaccessible habitat, e.g., from a cave stream or spring outlet to the aquifer that feeds it, capture-recapture data may indicate immigration to or emigration from these areas (*Means & Johnson, 1995*; *Day, Gerken & Adams, 2016*). Explicitly modeling transitions between observable and unobservable states using capture-recapture data (so-called temporary emigration) allows for estimates of demographic

parameters (*Kendall, Nichols & Hines, 1997*), breeding status (e.g., *Kendall & Bjorkland, 2001*), habitat use (*Bailey, Simons & Pollock, 2004a*; *Cecala, Price & Dorcas, 2013*), and movement (*Băncilă et al., 2018*). A temporary emigration model provides estimates of movement into and out of the sample area based on the transition rate between observable and unobservable states (*Kendall, Nichols & Hines, 1997*). In cases where observable states correspond to accessible areas (e.g., within a cave gallery or at a spring outlet), quantifying temporary emigration allows inferences about the characteristics of the population that use both the accessible and inaccessible areas rather than being limited to only the area directly sampled.

Barton Springs salamanders (*Eurycea sosorum*) occur in both surface and subterranean habitat, but populations are typically only accessible for study near springs. When first described, the salamanders were assumed to be restricted to the immediate vicinity of Barton Springs in Austin, Texas (*Chippindale, Price & Hillis, 1993*), one of several large spring complexes in the Edwards Aquifer of central Texas that harbors endemic species (*Bowles & Arsuffi, 1993*; *Krejca & Reddell, 2019*). Because the subterranean conduits feeding Barton Springs are largely inaccessible, biologists questioned whether the species was primarily a surface dweller that occasionally enters subterranean habitat or an obligate subterranean species that is flushed to the surface accidentally (*Sweet, 1978*; *Chippindale, Price & Hillis, 1993*). Following its description, in 1997 this species was listed as endangered under the US Endangered Species Act of 1973 due to its exceptionally small range (at the time, known from only three springs in a city park) and presumed low abundances (*Chippindale, Price & Hillis, 1993*; *City of Austin, 1997*; *US Fish and Wildlife Service, 1997*). Although additional occurrences of this species have subsequently been recorded outside of the type locality (*Bendik et al., 2013a*; *McDermid, Sprouse & Krejca, 2015*; *Devitt & Nissen, 2018*), abundances at those sites are lower and surface habitat is more limited in extent compared to Barton Springs. Abundances can be high at the type locality (e.g., >500 individuals) but may fluctuate by orders of magnitude and occasionally reach zero, or nearly so (*Bendik & Dries, 2018*; *Dries & Colucci, 2018*). This suggests that either surface populations are occasionally extirpated and recolonized from subterranean populations or that emigration from the surface to the aquifer is occurring (*Bendik & Dries, 2018*). Complete extirpation without emigration into the aquifer would suggest that the surface habitat is a population sink (i.e., unable to be maintained without immigration; *Pulliam, 1988*) and that surface populations could be a dead-end in terms of species persistence. If so, changes to population management practices and recovery planning may be necessary, because a primary focus of conservation for Barton Springs salamanders includes management and restoration of the surface habitat (*Dries et al., 2013*; *US Fish and Wildlife Service, 2016*).

Here, we report the results of a four-year robust-design capture-recapture study to better understand movement patterns and subterranean habitat use of Barton Springs salamanders. We examined how extrinsic environmental factors (spring discharge, turbidity, dissolved oxygen, temperature) and intrinsic factors (body size) affect survival and temporary emigration. Population dynamics differ between juveniles and adults in Barton Springs salamanders (*Bendik & Dries, 2018*), and size is often correlated with the propensity for movement in amphibians (*Cayuela et al., 2020*). Similarly, spring flow is

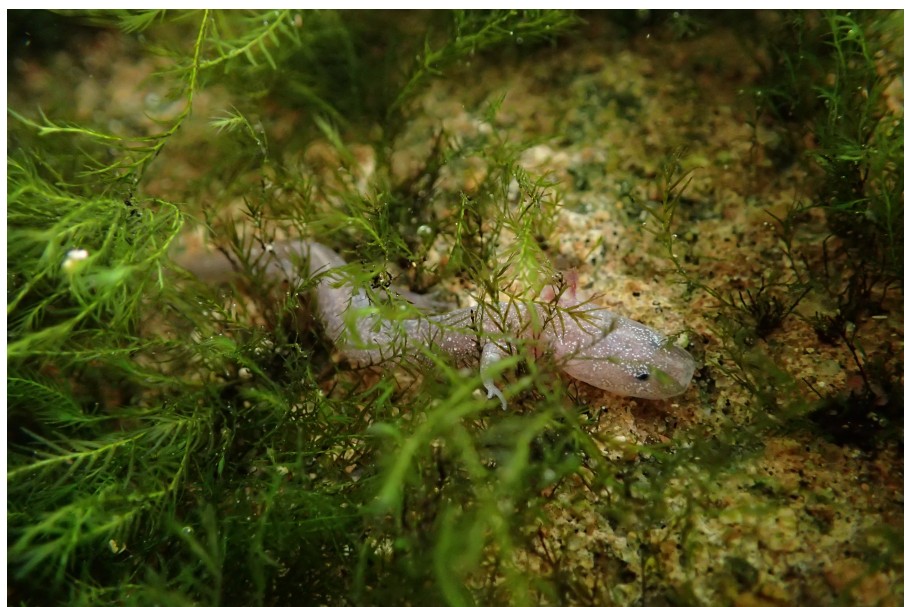

**Figure 1** **An adult Barton Springs salamander (*Eurycea sosorum*) moving amongst moss in Eliza Spring.** Photo Credit: Nathan Bendik, City of Austin.

an important driver of environmental conditions (e.g., *Mahler & Bourgeais, 2013*) that are important to the ecology and physiology of groundwater *Eurycea* salamanders (*Fries, 2002*; *Woods et al., 2010*; *Crow et al., 2016*; *Bendik & Dries, 2018*) and may influence movement patterns as well.

## MATERIALS & METHODS

### Study site

We studied Barton Springs salamanders (Fig. 1) at Eliza Spring (Fig. 2), part of the Barton Springs complex in Austin, Texas, USA. Eliza Spring is adjacent to Barton Springs Pool, a popular spring-fed swimming pool. Barton Springs is the primary discharge point for the Barton Springs segment of the Edwards Aquifer, a sole-source karstic aquifer that responds rapidly to changes in precipitation via a system of sinking streams that occur over a 430 km$^2$ area (*Hunt, Smith & Hauwert, 2019*). Eliza Spring emerges into a concrete-bottomed pool (74 m$^2$) with a water depth of approximately 0.3 m (Fig. 2A) and harbors the highest average densities of Barton Springs salamanders among known spring localities (4.32/m$^2$ ± 3.4 SD compared to ≤ 0.25/m$^2$ ± 0.3 at other sites within the Barton Springs complex; *Dries & Colucci, 2018*). Throughout most of our study, the spring pool discharged directly into a concrete pipe (0.6 m diameter) that flowed to Barton Creek, likely representing a point of no return for salamanders. During the last year of our study, the pipe was replaced by a 1 m wide overland stream that created 25 m$^2$ of additional habitat, but that still terminated at a concrete tunnel that conveys Barton Creek (Fig. 2B).

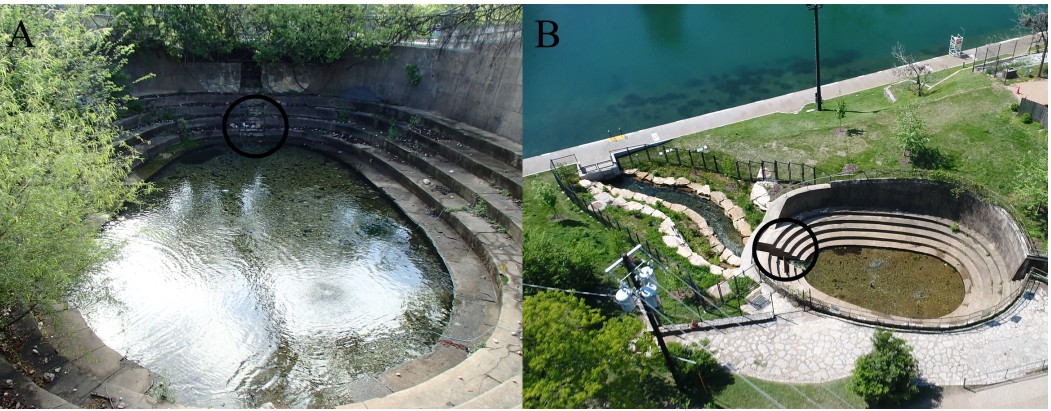

**Figure 2  Eliza Spring before (A) and after (B) overland stream reconstruction.** The black circle indicates the location of the outflow drain (A) that was daylighted (B). The new overland stream (B) now flows into the bypass tunnel (underneath the sidewalk) alongside Barton Springs Pool. Photo Credit: City of Austin.

## Data collection

We used a robust-design sampling scheme (*Pollock, 1982*) that consisted of 15 primary sampling events separated by ca. 3 month (range = 2–6 mo.) intervals during which the population was assumed to be open to births, deaths, and migration. During each primary sampling event, the population was sampled three times over the course of one week. We assumed the population was closed within each primary sampling event. We performed surveys from October 2014 through November 2018. While observers varied from survey to survey (typically a team of 4–6 people), a core group of 3–4 experienced observers were present during most of our sampling events. All fieldwork was performed under the authority of Texas Parks & Wildlife scientific permit SPR-0113-006 and US Fish and Wildlife permit TE-839031.

During surveys, biologists snorkeled from downstream to upstream, searching for salamanders by visually inspecting and moving all rocky substrate and attempting to capture all observed salamanders using aquarium nets. Salamanders were then temporarily held in flow-through mesh containers in the spring to await processing. Captured salamanders were photographed in a transparent water-filled tray against a five mm grid using a Nikon DSLR and macro lens with two flashes mounted on a custom stand.

We measured the body length for all individuals captured to the nearest 0.1 mm using ImageJ (*Rasband, 1997*). Body length was quantified as the mid-vertebral distance from the tip of the snout to the posterior insertion of the hindlimbs. Photographs were then cropped to include just the head, and we used program Wild-ID to identify individuals (*Bolger et al., 2012*) based on differences in color pattern. We used the R statistical software environment (*R Core Team, 2020*) to generate a matrix of capture histories from Wild-ID output. Photographic identification using this approach has been shown to be more accurate than physical tagging in a similarly patterned species (*E. tonkawae*; *Bendik et al., 2013b*). One limitation is that small juveniles cannot be tracked reliably using photographs

over periods greater than ca. two months because of changes in pigmentation during growth (*Bendik et al., 2013b*). For this reason, we excluded individuals <15.01 mm in body length (corresponding to ca. 25 mm in total length) from the capture-recapture analysis. The resulting data set included large juveniles and all adults, based on estimates of a mature size of 22.5 mm snout-vent length (*Chippindale, Price & Hillis, 1993*). Photographs in the final data set were visually matched and confirmed as a match or non-match using all 20 candidate matches presented by Wild-ID. To reduce visual matching errors, the photos were matched independently by two people and any conflicts found were investigated and resolved.

## Analysis

We estimated capture ($p$) and recapture ($c$) probabilities, survival ($S$), temporary emigration ($\gamma$) and abundance ($N$) using robust-design capture-recapture models. Capture probability is conditional upon the individuals being available for capture. Temporary emigration refers to the state when individuals in a population are temporarily unavailable for capture and is Markovian when movement from the study is dependent upon the prior state (*Kendall, Nichols & Hines, 1997*). Temporary movement of a salamander out of the sampling area, given that it was present during the previous period is estimated by the parameter $\gamma''$. Staying out of the sampling area given that it was not present during the previous period is $\gamma'$. When these parameters are equal, temporary movement is considered random. All temporary migrants are part of the superpopulation—the total population of individuals in the sampling area including those temporarily unavailable for capture and those available for capture (*Kendall, Nichols & Hines, 1997*). The survival parameter $S$ is a function of both mortality and permanent emigration. Survival is apparent because the fate of each animal that permanently leaves the study area cannot be known without auxiliary information (e.g., from radio telemetry), and thus mortality and permanent emigration are confounded. Abundance ($N$) for each period refers to the size of the population available at the surface for capture during sampling. We did not attempt to estimate the size of the superpopulation because this requires data (or assumptions) about the source of new recruits when temporary emigration is Markovian (*Wen et al., 2011*; *Kendall, Nichols & Hines, 1997*).

By quantifying temporary emigration, we can measure the prevalence of subterranean movement of Barton Springs salamanders at Eliza Spring. The bottom of the Eliza Spring pool was filled with concrete in the early 1900s, except for 22 formed outlets (7 in the floor and 15 on the sides) to convey the spring water, so most of the base-level substrate is impenetrable by salamanders except through those outlets. All migrants that eventually return to the surface must have retreated below ground, and this rate can be estimated as temporary emigration using capture-recapture data.

Because capture-recapture model sets with multiple parameters can result in hundreds or even thousands of different models when all possible combinations of all parameters and covariates are used, we built our model sets using a stage-based, step-down approach, starting with the most general model of time-variation supported by our data (*Lebreton et al., 1992*). This most general model let $S$ vary by year (ending in Oct./Nov.) and $\gamma''$ and

$\gamma'$ vary by period. We were unable to successfully fit models including full-time variation of $S$. Proceeding with this structure, we first fit a variety of models (without covariates on survival or emigration parameters) to determine the most parsimonious structure for $p$ and $c$, which included time variation within periods but equal across periods, variation between periods but constant within, full-time variation (different estimates for each individual survey), constant detection, and models where $p = c$ (and aforementioned time structures), for a total of $n = 20$ model structures of detection.

In the final model-building stage, we compared models where temporary emigration ($\gamma''$ and $\gamma'$) varied over periods or as a function of environmental or individual covariates and models where survival probabilities varied with these same covariates. Specifically, we used five covariates that describe aspects of water quality and quantity measured at Barton Springs: flow (spring discharge rate), change in flow between sampling periods, water temperature, dissolved oxygen (DO) concentration, and turbidity. Increases in flow may help flush additional animals to the surface via drift, but flow may also prompt compensatory movement back into the spring outlets, influencing the rate of temporary migration in either direction. Permanent migration in response to discharge may also be reflected by apparent survival estimates. In general, increasing flow is thought to be beneficial to Barton Springs salamanders, particularly when recovering from drought conditions (*Dries & Colucci, 2018*). Our expectation is that spring flow may have a positive influence on actual survival, but high flow could prompt migration underground, producing an overall negative effect on apparent survival if this migration is permanent. Given the complexity of possible responses, we did not have a strong a priori hypothesis for the direction of the effect of these discharge covariates. Low DO and high temperature negatively affect survival and growth of Barton Springs salamanders under laboratory conditions (*Woods et al., 2010*; *Crow et al., 2016*), but these conditions infrequently occur in the wild, so we generally expect DO and temperature to have a positive influence on survival. Periods of high turbidity may result in excess fine sediment accumulation within salamander habitat, which has a negative relationship with relative abundance (*Bendik & Dries, 2018*). We therefore expected turbidity to have a negative relationship with survival and the probability of remaining above ground.

We summarized mean daily water quality and water quantity statistics for Barton Springs from United States Geological Survey (USGS) gage number 08155500. Mean spring flow, temperature, DO and turbidity were calculated from the mean daily statistics between each period (Table 1). Change in flow was calculated using the difference between the first and last day of the interval between surveys (Table 1). All covariates were z-scored prior to analysis. Rather than include all possible combinations of covariate parameterizations in our model set, we chose the three sets of covariates to evaluate as a function of survival and/or temporary emigration: "flow-only" (mean flow + change in flow), "flow-temp-turbidity" (mean flow + change in flow + mean temp + mean turbidity), and "turb-DO" (turbidity + DO). Because water temperature and flow can be correlated with DO, we excluded those combinations (e.g., flow and DO: $\rho = 0.74$, $P = 0.003$). We did not find problems with multicollinearity among covariates based on a condition number test. We

**Table 1  Summary of environmental variables measured at Barton Springs from USGS site 08155500.** Means were computed from daily statistics for each sampling interval. The grand mean and pooled range are summary statistics for the sampling interval means (i.e., the covariate values used). The total range represents the range of all daily statistics.

| Variable | Grand Mean ± SD | Pooled range | Total range |
|---|---|---|---|
| Flow (ft³/s) | 86.3 ±26.5 | 41–123 | 26–131 |
| Change in flow (ft³/s) | 4.05 ±27.1 | −31.3–2.8 | NA |
| Dissolved oxygen (mg/L) | 6.06 ±0.366 | 5.38–6.64 | 4.80–7.50 |
| Turbidity (FNU) | 2.30 ±0.510 | 1.72–2.52 | 0.8–31.4 |
| Temperature (°C) | 21.2 ±0.453 | 20.4–21.9 | 18.6–22.7 |

tested all combinations of the specified covariate and non-covariate models for survival and temporary emigration for a resulting model set of $n = 80$.

We also included individual covariates of body size on survival and temporary emigration. We expected survival to be lower for juvenile salamanders compared to adults (*Lee et al., 2012*; *Messerman, Semlitsch & Leal, 2020*). Furthermore, lower swimming performance in smaller salamanders (e.g., in *Eurycea bislineata*, *Azizi & Landberg, 2002*) may inhibit them from migrating back through flowing spring orifices, thus limiting their potential for temporary emigration away from the study site. Because size cannot be observed when individuals are not captured, we used a von Bertalanffy growth model (*Eaton & Link, 2011*) to predict body lengths for unobserved individuals following first capture. Several parameters are required to describe a von Bertalanffy growth function: initial size ($s[0]$), asymptotic size ($a$), and a growth rate coefficient ($k$). An additional parameter, $\lambda$, represents individual heterogeneity in the growth curves as the mean to variance ratio (*Eaton & Link, 2011*). Using Bayesian analysis in JAGS (*Plummer, 2003*), we estimated parameters $a$, $k$, $\lambda$ as well as standard deviation of measurement error ($sdme$) from body size measurements of up to five between-period recaptures, including individuals below our size cutoff. Thus, we took advantage of all the recapture information from smaller individuals to estimate the growth curve, even though some of these were excluded from the CMR analysis. Four chains of 100,000 iterations were run following a burn-in of 30,000, and convergence was confirmed via examination of trace plots and *Gelman & Rubin*'s (*1992*) diagnostic. We used the posterior means of $a$ and $k$, and the initial size, $s[0] = 7.53$ mm (based on observations of captured hatchlings) to predict sizes. Size covariates were coded as categorical variables based on the mean size at first capture among recaptured individuals (24.2 mm). This allowed for an even distribution among recaptures that would transition to a larger size. Individuals >24.2 mm were coded as 1, all others as 0, and transitions from size 1 to size 0 were not permitted. We compared all combinations of models for survival and temporary emigration parameters with body size as a single covariate or as additive with time ($n = 26$).

Our approach assumes transitions between states are estimated without error and we did not attempt to incorporate uncertainty in predictions of size. Although transition rates between states can be estimated with multi-state capture-recapture models (*Lebreton et al.,*

*2009*), we view the growth model approach as more accurate because it directly incorporates information (initial size, elapsed time between surveys, shape of the growth curve) about the transition (growth) process. To assess the sensitivity of this approach to variation in the growth rate parameters, we performed a bootstrapping procedure with 1,000 iterations. For each iteration, we randomly sampled parameter estimates of $a$ and $k$ from a single posterior MCMC draw. Next, we estimated size classes for all individuals after first capture, if unobserved. We then performed model fitting for the top model with size as a covariate on $S$, $\gamma''$ and/or $\gamma'$. Finally, we calculated Akaike Information Criterion corrected for small samples (AICc) to select the best models (*Burnham & Anderson, 2002a*).

We fit the data using maximum likelihood under the Huggins formulation (*Huggins, 1989*; *Huggins, 1991*), as implemented in program MARK v 9.0 (*White, 2020*) and used AICc and AICc weights ($w$) to compare the relative strength of each model (*Burnham & Anderson, 2002a*). We used the package RMark v 2.2.7 (*Laake, 2013*) implemented in R v3.6 and v4.0 (*R Core Team, 2020*) to build models and generate model-averaged parameter estimates. Standard errors for the average of time-varying parameters were calculated using the delta method.

## RESULTS

Our capture-recapture data set included 2,046 observations of 1,578 Barton Springs salamanders, 333 of which were recaptured at least once between periods. Almost all salamander observations occurred within the spring pool (99%) and only 1% occurred within the newly constructed stream (the stream was not fully colonized yet; unpublished data). During the first stage of model fitting, the best model of detection was $p = c$ with full-time variation across all survey events ($w$ =1.00). Model-averaged values of $\hat{p}$ ranged from 0.25 (SE = 0.054) during the second survey of November 2019 to 0.82 (SE = 0.043) during the first survey of November 2017. The most general models with temporary emigration were ranked higher than the non-movement model ($\Delta$AICc = 1065), with Markovian movement favored over the random movement model ($w = 1.00$; $\Delta$AICc = 42.4). None of the models including environmental covariates were favored (Table 2; sum of $w = 0$). In fact, our results indicated the effects of environmental variables (DO, spring flow, turbidity) were far outweighed by the effect of body size as a determinant of surface availability and survival (Table 2). Models with time-varying survival and emigration parameters were also more parsimonious than those where temporal variation was explained by environmental covariates (Table 2). The best model overall was $S$ (~size), $\gamma''$ (~time + size), $\gamma'$ (~time), $p = c$ (~period:time) ($w = 0.35$; Table 2). Goodness-of-fit tests are not available for robust-design models, but we note that the order of the top models was invariant to adjustments of the variance inflation factor, $\hat{c}$.

Despite models with environmental covariates not ranking highly, a comparison among models ($n = 27$) that only included environmental covariates as a function of survival and emigration parameters provides some insight as to the direction of different environmental effects (Table 3). An exhaustive model set of all possible covariate combinations (which is generally not recommended; *Burnham & Anderson, 2002b*) precludes use of model

**Table 2  Model selection results based on AICc for the top 15 capture-recapture models of survival, temporary emigration, and capture probability.**

| Model | AICc | ΔAICc | AICc weights ($w$) | Num. Par | -2logLik |
|---|---|---|---|---|---|
| $S$ (~size) $\gamma''$ (~time + size) $\gamma'$ (~time) | 10654.86 | 0.00 | 0.35 | 75 | 10501.47 |
| $S$ (~size) $\gamma''$ (~time + size) $\gamma'$ (~time + size) | 10655.12 | 0.26 | 0.31 | 76 | 10499.63 |
| $S$ (~year + size) $\gamma''$ (~time + size) $\gamma'$ (~time) | 10655.71 | 0.85 | 0.23 | 78 | 10496.04 |
| $S$ (~year + size) $\gamma''$ (~time + size) $\gamma'$ (~time + size) | 10657.10 | 2.24 | 0.11 | 79 | 10495.33 |
| $S$ (~year + size) $\gamma''$ (~time + size) $\gamma'$ (~size) | 10680.08 | 25.21 | 0.00 | 67 | 10543.37 |
| $S$ (~year + size) $\gamma''$ (~time) $\gamma'$ (~time) | 10680.48 | 25.61 | 0.00 | 77 | 10522.90 |
| $S$ (~year + size) $\gamma''$ (~time) $\gamma'$ (~time + size) | 10682.57 | 27.70 | 0.00 | 78 | 10522.89 |
| $S$ (~size) $\gamma''$ (~time) $\gamma'$(~time) | 10685.86 | 31.00 | 0.00 | 74 | 10534.56 |
| $S$ (~size) $\gamma''$ (~time + size) $\gamma'$ (~size) | 10687.29 | 32.42 | 0.00 | 64 | 10556.82 |
| $S$ (~size) $\gamma''$ (~time) $\gamma'$ (~time + size) | 10687.85 | 32.98 | 0.00 | 75 | 10534.45 |
| $S$ (~year) $\gamma''$ (~time + size) $\gamma'$ (~time) | 10692.17 | 37.30 | 0.00 | 77 | 10534.59 |
| $S$ (~flow-temp-turbidity) $\gamma''$ (~time) $\gamma'$ (~time) | 10693.31 | 38.44 | 0.00 | 77 | 11715.30 |
| $S$ (~year) $\gamma''$ (~time + size) $\gamma'$ (~time + size) | 10694.03 | 39.17 | 0.00 | 78 | 10534.35 |
| $S$ (~flow-only) $\gamma''$ (~time) $\gamma'$ (~time) | 10694.64 | 39.78 | 0.00 | 75 | 11720.82 |
| $S$ (~turb-DO) $\gamma''$ (~time) $\gamma'$ (~time) | 10695.45 | 40.59 | 0.00 | 75 | 11721.63 |

**Notes.**

S, apparent survival; $\gamma''$, unavailable at the surface | present during the last survey; $\gamma'$, unavailable at the surface | unavailable during the last survey.

All models shown below include $p = c$ within and between session variation for capture and recapture probabilities.

averaging of the beta estimates. Instead, we provide coefficients (on the logit scale) from the top five models (sum of AICc $w = 0.88$) within this subset to illustrate the direction of environmental effects (Table 4). Estimates for the top models were generally consistent and did not change sign (Table 4). Flow-temp-turbidity and flow-only models of survival outperformed turb-DO models, while turb-DO models for emigration were favored over others (Table 3). Temperature and flow were positively correlated with survival, which was consistent with our hypothesis (Table 4). Turbidity was positively associated with both emigration parameters, but only significantly so for $\gamma''$. DO was negatively associated with both emigration parameters, but again, this affect was negligible for $\gamma'$. The remaining coefficients had 95% confidence intervals overlapping zero (Table 4). Collectively, these results suggest that salamanders move below ground with increasing turbidity and decreasing DO, which was consistent with our expectation.

Growth was modeled from 901 measurements of 372 recaptured individuals. Mean parameter estimates with 95% credence intervals from the von Bertalanffy growth model were as follows: $a = 31.57$ (30.99–32.18), $k = 7.81$ (7.14–8.52), $\lambda = 4.49$ (3.34–6.01) and $sdme = 1.25$ (1.11–1.39). The bootstrap analysis using random draws from the posterior distributions of parameter estimates from the growth model showed little variation in AICc for the top model (range 10654.32–10655.75), indicating that different potential realizations of the individual growth trajectories (based on the population mean) have little impact on our inference. This is due to our study design sampling intervals being long enough and growth of salamanders fast enough (Fig. 3), that small shifts in growth rate

**Table 3** Model selection results based on AICc for the top subset of models comparing environmental effects on survival and temporary emigration.

| Model | AICc | ΔAICc | AICc weights ($w$) | Num. Par | Deviance |
|---|---|---|---|---|---|
| $S$ (flow-temp-turbidity) $\gamma''$ (turbidity-DO) $\gamma'$ (turbidity-DO) | 10749.22 | 0 | 0.38 | 56 | 11814.91 |
| $S$ (flow-only) $\gamma''$ (turbidity-DO) $\gamma'$ (turbidity-DO) | 10750.43 | 1.21 | 0.21 | 54 | 11820.25 |
| $S$ (flow-temp-turbidity) $\gamma''$ (turbidity-DO) $\gamma'$ (flow-temp-turbidity) | 10751.59 | 2.36 | 0.12 | 58 | 11813.13 |
| $S$ (flow-only) $\gamma''$ (turbidity-DO) $\gamma'$ (flow-only) | 10751.92 | 2.69 | 0.10 | 54 | 11821.74 |
| $S$ (flow-temp-turbidity) $\gamma''$ (turbidity-DO) $\gamma'$ (flow-only) | 10752.11 | 2.89 | 0.09 | 56 | 11817.80 |
| $S$ (flow-only) $\gamma''$ (turbidity-DO) $\gamma'$ (flow-temp-turbidity) | 10753.53 | 4.31 | 0.04 | 56 | 11819.22 |
| $S$ (flow-temp-turbidity) $\gamma''$ (flow-temp-turbidity) $\gamma'$ (turbidity-DO) | 10754.79 | 5.57 | 0.02 | 58 | 11816.34 |
| $S$ (flow-only) $\gamma''$ (flow-temp-turbidity) $\gamma'$ (turbidity-DO) | 10755.53 | 6.30 | 0.02 | 56 | 11821.22 |
| $S$ (flow-temp-turbidity) $\gamma''$ (flow-temp-turbidity) $\gamma'$ (flow-temp-turbidity) | 10757.31 | 8.09 | 0.01 | 60 | 11814.72 |
| $S$ (flow-only) $\gamma''$ (flow-temp-turbidity) $\gamma'$ (flow-only) | 10757.37 | 8.15 | 0.01 | 56 | 11823.06 |

Notes.

S, apparent survival; $\gamma''$, unavailable at the surface | present during the last survey; $\gamma'$, unavailable at the surface | unavailable during the last survey.

All models shown below include $p = c$ within and between session variation for capture and recapture probabilities. ΔAICc values are only relative to this subset of models.

**Table 4** Coefficients representing environmental effects from the subset of models comparing environmental effects on survival and temporary emigration.

| | Model sub-rank | | | | |
|---|---|---|---|---|---|
| Parameter, covariate | 1 | 2 | 3 | 4 | 5 |
| $S$, temperature | **0.35 (0.03,0.67)** | | **0.39 (0.06,0.72)** | | 0.30 (−0.02,0.63) |
| $S$, spring flow | **0.39 (0.15,0.64)** | **0.41 (0.21,0.62)** | **0.36 (0.06,0.66)** | **0.43 (0.18,0.67)** | **0.42 (0.15,0.69)** |
| $S$, change in flow | 0.27 (−0.14,0.68) | 0.13 (−0.11,0.36) | 0.28 (−0.15,0.71) | 0.13 (−0.10,0.37) | 0.28 (−0.13,0.69) |
| $S$, turbidity | 0.10 (−0.33,0.54) | | 0.07 (−0.38,0.52) | | 0.08 (−0.32,0.49) |
| $\gamma''$, turbidity | **0.54 (0.35,0.73)** | **0.49 (0.31,0.68)** | **0.52 (0.32,0.72)** | **0.49 (0.30,0.68)** | **0.54 (0.34,0.74)** |
| $\gamma''$, dissolved oxygen | **−0.38 (−0.56,-0.20)** | **−0.32 (−0.49,-0.14)** | **−0.40 (−0.60,-0.21)** | **−0.31 (−0.50,-0.12)** | **−0.36 (−0.56,-0.16)** |
| $\gamma'$, turbidity | 0.15 (−0.07,0.36) | 0.10 (−0.12,0.31) | 0.06 (−0.29,0.41) | | |
| $\gamma'$, dissolved oxygen | −0.23 (−0.52,0.06) | −0.17 (−0.44,0.09) | | | |
| $\gamma'$, spring flow | | | −0.25 (−0.61,0.11) | −0.13 (−0.44,0.19) | −0.15 (−0.49,0.19) |
| $\gamma'$, change in flow | | | 0.05 (−0.31,0.41) | 0.07 (−0.17,0.31) | 0.11 (−0.12,0.34) |
| $\gamma'$, temperature | | | 0.20 (−0.01,0.41) | | |

Notes.

Estimates (and 95% confidence intervals) on the logit-scale are represented in boldface if the confidence interval did not contain zero. Values are given for the top five models within this subset (sum of AICc w = 0.88).

parameters have a negligible effect on the predicted transition rates between size classes at the resolution of our data.

Body size had a positive effect on $S$ and $\gamma''$. Estimates of the body size effect on the logit scale (beta) from the top model were 1.43 (SE = 0.17) and 1.45 (SE = 0.28) for $S$ and $\gamma''$, respectively. Among both size classes, estimates of monthly apparent survival, $\hat{S}$, ranged between 0.75 (SE = 0.04) and 0.95 (SE = 0.02) (Fig. 4A), which corresponds to

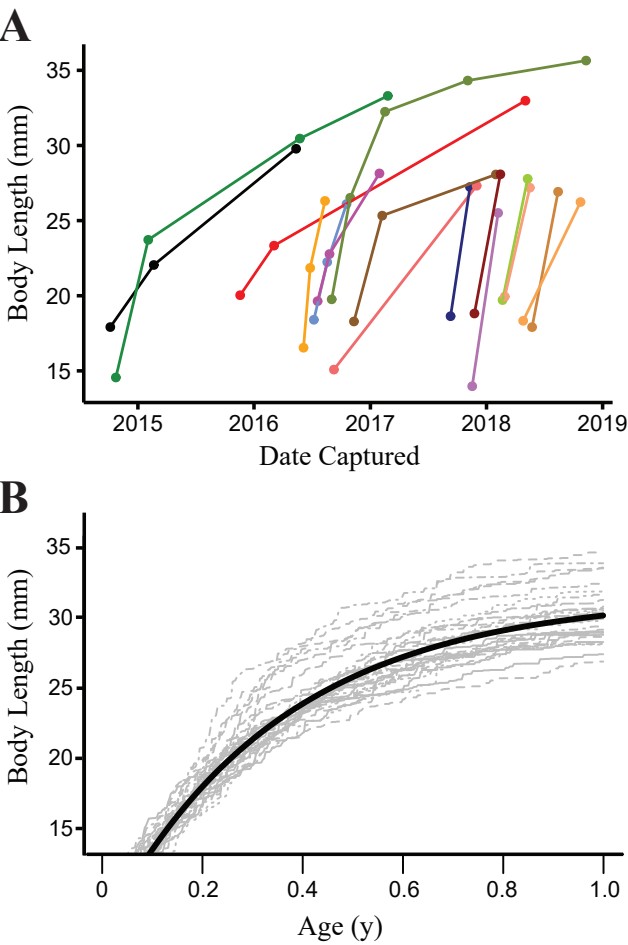

**Figure 3  Growth of Barton Springs salamanders.** (A) Measured growth in body length of 16 Barton Springs salamanders at different capture intervals. (B) Expected size for salamanders > 14 mm body length during the first year of growth based on a von Bertalanffy growth model. The dark solid line indicates age-at-length estimated from the mean parameter values for *a* and *k*. Light gray lines are from 30 randomly generated growth curves from the mean parameter estimates to demonstrate individual variation.

annual survival rates between 0.03 and 0.52. Temporary emigration varied among size classes ($\widehat{\gamma''}$ range: 0.16–0.97); the probability of moving from the pool to subterranean habitat was higher for larger individuals (mean $\widehat{\gamma''} = 0.79$, SE = 0.02) compared to smaller individuals (mean $\widehat{\gamma''} = 0.53$, SE = 0.06) (Fig. 4). Similarly, the probability of remaining in subterranean habitat ($\widehat{\gamma'}$ range: 0.59–0.96) was higher for larger individuals (mean $\widehat{\gamma'} = 0.86$, SE = 0.02) than smaller individuals (mean $\widehat{\gamma'} = 0.77$, SE = 0.17) (Fig. 4). Abundance ($\hat{N}$) at the surface during each primary period ranged from 62 (SE = 3) to 377 (SE = 9) (Fig. 5).

A typical recaptured individual was not observed for at least one survey period before being observed again, which is consistent with the high estimates of temporary emigration. The median time for between-period recaptures was 105 days, which is the average time

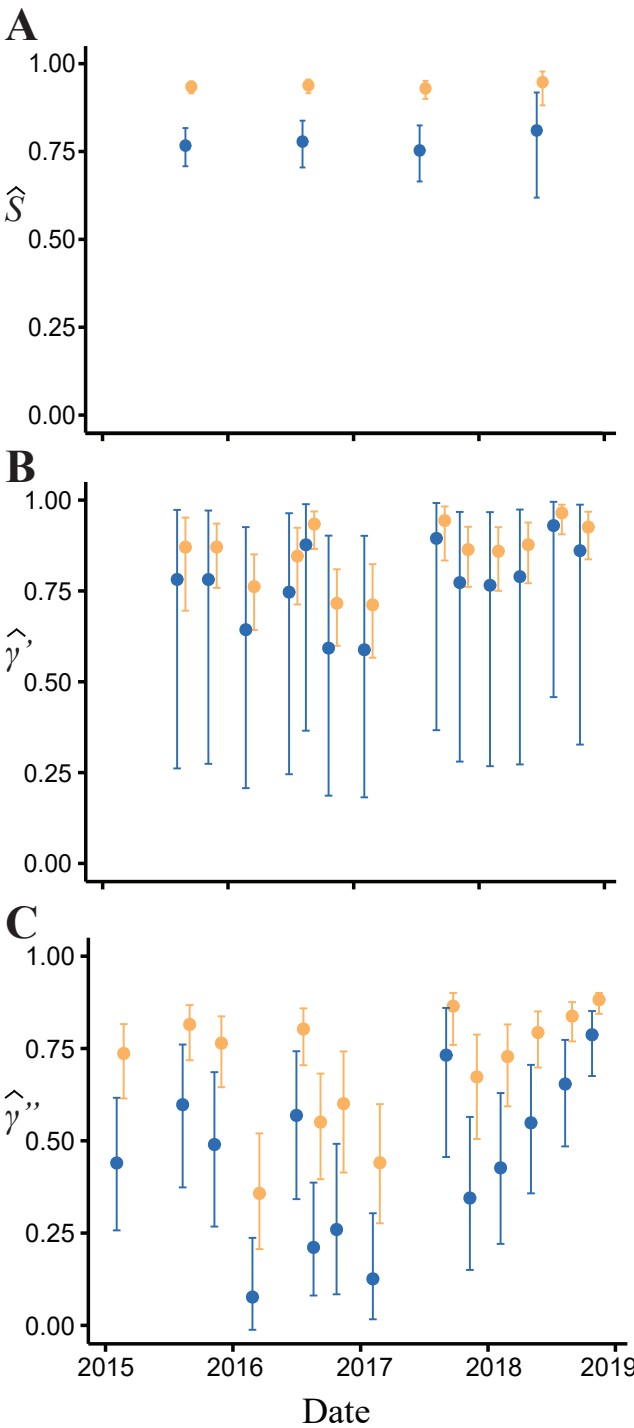

**Figure 4 Model-averaged estimates of monthly apparent survival and temporary emigration.** (A) Apparent survival (B) $\gamma'$ = unavailable at the surface | unavailable during the last survey (C) $\gamma''$ = unavailable at the surface | present during the last survey. Dark blue indicates estimates for juvenile and small adults (15.01–24.20 mm body length); light orange indicates large adults ( > 24.20 mm).

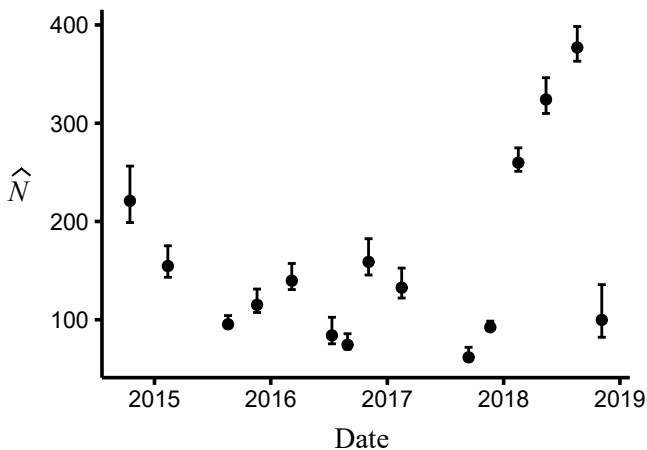

**Figure 5** **Abundance of Barton Springs salamanders at the surface of Eliza Spring from October, 2014 through November, 2018.** Error bars represent 95% confidence intervals.



interval between surveys. In some cases, individuals were not observed for several survey periods before being recaptured. For example, 20% of between-period recaptures (66 of 333) were not seen for more than one year at some point during the study, and 13 individuals had at least a two-year span between observations.

## DISCUSSION

Our results contradict characterizations of Barton Springs salamanders as primarily surface-dwellers, or as subterranean inhabitants accidentally flushed to the surface (see, e.g., *Sweet, 1978*; *Chippindale, Price & Hillis, 1993*). Barton Springs salamanders at Eliza Spring can be found in abundance at the surface (*Bendik & Dries, 2018*; *Dries & Colucci, 2018*), but often move underground. High estimates of $\gamma''$ demonstrate that temporary emigration of individuals from the surface to the subsurface is frequent. A large proportion of individuals also remained underground between surveys, as shown by similarly high estimates of $\gamma'$. Some individuals were not recaptured until more than a year later, suggesting that salamanders may spend long periods of time below ground before returning to the surface. Movements between the surface and subsurface may be a large component of previously observed surface population dynamics, which are characterized by high variability in abundance (*Bendik & Dries, 2018*; *Dries & Colucci, 2018*).

High temporary emigration in Barton Springs salamanders is consistent with observations of other plethodontid salamander populations (*Bailey, Simons & Pollock, 2004a*; *Bailey, Simons & Pollock, 2004b*; *Price, Browne & Dorcas, 2012*; *O'Donnell & Semlitsch, 2015*; *Bendik, 2017*; *Drukker et al., 2018*). In general, surface activity of plethodontid salamanders tends to be low, with surface abundance being a small fraction of the superpopulation size (e.g., see Table 1 in *O'Donnell & Semlitsch, 2015*). Plethodontids migrate beneath the forest floor (*Bailey, Simons & Pollock, 2004a*; *O'Donnell, Thompson & Semlitsch, 2014*) or to aquatic subterranean refugia (*Bonett & Chippindale, 2006*; *Bendik & Gluesenkamp, 2013*; *Steffen et al., 2014*) in response to moisture conditions at the surface,
which can influence apparent demographic patterns (*Connette, Crawford & Peterman, 2015*; *O'Donnell & Semlitsch, 2015*). Although temporary emigration in plethodontids is common, reasons for lower surface availability of Barton Springs salamanders are likely different than those terrestrial salamanders. Lungless salamanders must keep their skin moist for cutaneous respiration and to prevent water loss (*Feder, 1983*) which influences their surface activity. Neotenic *Eurycea* require suitable levels of DO (*Woods et al., 2010*) and flowing water to maintain activity at the surface (*Bendik & Gluesenkamp, 2013*), but otherwise should be unaffected by the challenges of variable moisture faced by their terrestrial counterparts.

Models with environmental effects performed poorly compared to models with individual effects (body size) or time-varying parameters. Despite this, environmental effects can still provide some insight into the temporal variation we observed in salamander movement. Turbidity was positively associated with movement below ground, suggesting that either conditions worsen on the surface during higher turbidity or they improve below ground relative to the surface (e.g., via increased organic matter availability in the subterranean ecosystem). This pattern is consistent with observations of lower surface abundance when fine sediment deposition is high (*Bendik & Dries, 2018*). DO values were negatively associated with salamanders moving below ground but not significantly associated with movement to the surface. Salamanders may move in response to declining DO in anticipation of adverse conditions, for example, as values below 4.4 mg/L start to have negative effects on salamander growth and survival (*Woods et al., 2010*). However, this is speculative because DO remained above 4.8 mg/L during our study (Table 1). Furthermore, DO does not vary substantially between the near-surface and surface habitats and is higher at the springs than deep underground (*Mahler & Bourgeais, 2013*), so salamander movement below ground in search of higher DO conditions is unlikely. However, subterranean habitat could be less metabolically taxing for individuals (e.g., less movement and lower stress due to lack of predation) and prompt migration underground as DO declines. DO is correlated with flow, which was positively associated with survival and consistent with our hypothesis. Other studies have indicated the importance of this variable for Barton Springs salamander population demographics. For example, lagged flow is positively correlated with reproduction and low flows are associated with low abundance (*Bendik & Dries, 2018*; *Dries & Colucci, 2018*). Spring flow may also be associated with food availability within the aquifer, although this hypothesis is untested. Temperatures during our study were within the range for optimal growth for Barton Springs salamanders (15–24 °C; *Crow et al., 2016*), so response to temperature may be an indicator for other ecological conditions that improve survival, such as food availability, rather than indicating a direct effect on mortality. DO, temperature and flow are intertwined in this system and can exhibit correlated, but complex non-linear relationships (*Mahler & Bourgeais, 2013*); ultimately it is hard to isolate these effects. While an exhaustive post-hoc analysis of environmental covariates was beyond the scope of this study, this approach might be useful in the future to explore a suite of more complex models.

Smaller salamanders (<24 mm body length) had lower survival and a lower probability of moving underground, which was consistent with our prediction. Juvenile *Eurycea* have

difficulty swimming against strong currents (*Barrett et al., 2010*) and are weaker swimmers than larger individuals (*Azizi & Landberg, 2002*). The spring outlets at Eliza Spring can reach high velocities (mean = 0.16 m/s, SD = 0.16, max = 0.65; *City of Austin, 2020*) in a range that has been shown to flush juvenile *Eurycea cirrigera* from substrates without gravel (*Barrett et al., 2010*). If smaller salamanders have more difficulty seeking subterranean refuge at the spring surface, they will be less able to respond to increases in competition for space or predation pressure compared to larger individuals, which more readily move into subterranean refugia. This may, in part, explain why survival is also lower for juvenile salamanders, and could result in a pattern of negative density dependence, as observed in a time-series analysis from two-decades of monthly Barton Springs salamander counts (*Bendik & Dries, 2018*). If this were the case, we should expect survival to vary with both size and year, given the differences in abundance we observed over the course of our study. In our analysis, however, time-dependent models of survival carried less weight (sum of $w$ = 0.44) than those with size alone (sum of $w$ = 0.66). Furthermore, a post-hoc comparison of $S$ with an interaction between year and size did not improve upon our top model, suggesting that if there is a survival-density relationship on smaller salamanders, it is not apparent from these data. Negative density-dependence at the surface may also manifest itself through patterns of movement. There is a positive correlation between estimates of $\hat{N}$ at the beginning of the time interval and $\widehat{\gamma''}$ for that interval, and this relationship is stronger for the smaller size class (Pearson's product-moment correlation, $\rho = 0.60$, $P = 0.02$) compared to the larger size class ($\rho = 0.49$, $P = 0.08$). Therefore, salamanders may be moving underground in response to increased density at the surface, and there is evidence of density-dependent movement between habitats in other amphibians (*Grayson, Bailey & Wilbur, 2011*). Alternative statistical approaches to explore the relationship between density and demographic rates may be a useful avenue of future research (e.g., *Kissel, Tenan & Muths, 2020*).

Barton Springs salamanders may move into subterranean refugia for reproduction, or to avoid predation. Egg-laying occurs below ground and courtship may as well, and reproduction is not strictly seasonal as is the case for many other amphibians. Most individuals in the small size class were reproductively immature (85% were <22.5 mm in body length; *Chippindale, Price & Hillis, 1993*), which may partly explain why temporary emigration was lower for this group. Both predators and prey of Barton Springs salamanders are more abundant in surface habitat, but we did not measure variation in either of these factors for this study. Potential predators include crayfish (*Procambarus clarkii*), mosquitofish (*Gambusia affinis*) and larval damselflies (e.g., *Argia* spp.), while their prey includes a variety of invertebrates (*DeSantis, Davis & Gabor, 2013*; *Gillespie, 2013*; *Owen et al., 2016*; *Davis, De Santis & Gabor, 2017*). Below ground, the picture is quite different, where Barton Springs salamanders are likely the top predator (along with the sympatric *E. waterlooensis*), but invertebrate density is expected be much lower owing to a dearth of primary production in groundwater ecosystems (*Gibert, Danielopol & Stanford, 1994*). Barton Springs salamanders may therefore face a tradeoff between increased food availability at the surface vs. little to no interspecific predation in subterranean habitat. Growth is size-dependent and occurs rapidly during the first year of life (Fig. 3), so

salamanders may take advantage of the resource-rich environment at the surface and therefore spend more time above ground when young. However, more research is required to understand ecological consequences of surface and subterranean habitat for these salamanders. Unfortunately, the subterranean ecology of the Barton Springs segment of the Edwards Aquifer is largely undocumented; even basic research on the diversity and distribution of its fauna is scarce (but see *Hutchins, 2018*; *Nissen et al., 2018*), despite otherwise extensive study of its hydrogeology and geochemistry (*Hunt, Smith & Hauwert, 2019* and references therein).

## Conservation implications

The City of Austin has a Habitat Conservation Plan that includes numerous measures to improve the surface habitat for Barton Springs salamanders, such as sediment mitigation and habitat expansion (*Dries et al., 2013*). Given the limited extent to which conservation actions can be implemented at Barton Springs, we believe improvement of surface habitat continues to be a worthwhile goal. The high frequency of movement between the surface and subsurface indicates that the population at Eliza Spring is not a population sink maintained solely by immigration from the aquifer, and that a large proportion of the population associated with the surface is typically underground. Thus, conservation actions meant to improve population persistence at the surface therefore have the potential to improve persistence of the local resident population in and around Barton Springs. For example, a recent expansion of surface habitat at Eliza Spring occurred with the addition of an overland stream (Fig. 2B). Moving forward, we can build on results from this study to examine if carrying capacity or survival improve because of the newly expanded habitat. Furthermore, periodic declines in apparent surface abundance do not necessarily reflect an existential threat to the species, as was once thought. However, the extent to which local population persistence contributes to the conservation status of the species as a whole is uncertain. Newly documented localities for Barton Springs salamanders (*Devitt & Nissen, 2018*) expand the known range of this species by (potentially) hundreds of square km within the Barton Springs segment of the Edwards Aquifer and the adjacent Trinity Aquifer. One individual was captured deep within a well (*McDermid, Sprouse & Krejca, 2015*), while others have been regularly observed within a cave stream that eventually flows to Barton Springs (Blowing Sink Cave; *Bendik et al., 2013a*; *City of Austin, 2018a*). Data on abundance and density for populations outside of Barton Springs are scarce and difficult to obtain, but available evidence suggests that both population density and spatial density (number of sites/area) both appear to be lower outside of the Barton Springs complex (*Devitt & Nissen, 2018*; *Dries & Colucci, 2018*; *City of Austin, 2018a*). While this observation is partly biased by our inability to adequately explore subterranean habitats, it underscores the need to understand how populations at the endpoint of the Barton Springs system interact with other, lower density populations throughout their range. The direction and frequency of gene flow throughout the range of Barton Springs salamanders is of particular interest, because species persistence may hinge on the interconnectivity of these sparse populations (*City of Austin, 2018b*). However, because of limited access to

subterranean sites, inferences about population connectivity may have to rely on fortuitous sampling of difficult-to-access habitat.

## CONCLUSIONS

Since Barton Springs salamanders were described from a single spring locality in 1993, knowledge of their population ecology has been gleaned from surface count surveys, limiting any inference about what we now know is a fraction of a larger superpopulation (*City of Austin, 1997*; *US Fish and Wildlife Service, 2016*; *Bendik & Dries, 2018*; *Dries & Colucci, 2018*). Using an indirect approach to detect movement, we quantified temporary emigration of Barton Springs salamanders into subterranean habitat from capture-recapture data. Temporary emigration was frequent, indicating that apparent abundance at the surface is not solely a function of immigration, recruitment and/or survival. Furthermore, these findings suggest that periodic declines in surface abundance of Barton Springs salamanders at Barton Springs are not wholly caused by mortalities, but may include emigration to adjacent, unobservable habitat. Our results demonstrate the utility of using estimates of temporary emigration to understand movement and habitat use in species where direct observation is not always possible.

## ACKNOWLEDGEMENTS

We thank the following individuals for aiding with field surveys (in no particular order): Justin Crow, Pete Diaz, Andy Gluesenkamp, Romey Swanson, Kelsey Anderson, Radmon Rice, Donella Strom, Saj Zappitello, Crystal Datri, Lindsey Sydow, Scott Hiers, Jonny Scalise, Charlotte Kucera, Todd Jackson, Luis Medina, Liza Colucci, and Jesse Meik. Ron Bonett, Jesse Meik and two anonymous reviewers provided helpful comments on the manuscript. The findings and conclusions in this article are those of the author(s) and do not necessarily represent the views of the US Fish and Wildlife Service.

### Funding
This work was supported by the City of Austin.

### Grant Disclosures
The following grant information was disclosed by the authors:
City of Austin.

### Competing Interests
The authors declare there are no competing interests.

### Author Contributions
- Nathan F. Bendik conceived and designed the experiments, performed the experiments, analyzed the data, prepared figures and/or tables, authored or reviewed drafts of the paper, and approved the final draft.

- Dee Ann Chamberlain, Thomas J. Devitt and Donelle Robinson conceived and designed the experiments, performed the experiments, authored or reviewed drafts of the paper, and approved the final draft.
- Sarah E. Donelson, Bradley Nissen, Jacob D. Owen and Kenneth Sparks performed the experiments, analyzed the data, authored or reviewed drafts of the paper, and approved the final draft.
- Blake N. Sissel conceived and designed the experiments, performed the experiments, analyzed the data, authored or reviewed drafts of the paper, and approved the final draft.

## Animal Ethics

The following information was supplied relating to ethical approvals (i.e., approving body and any reference numbers):

All work has been approved and was performed under the authority of state and federal permits for endangered species: Texas Parks & Wildlife scientific permit SPR-0113-006 and U.S. Fish and Wildlife permit TE-839031. Work was also approved by an internal quality assurance committee.

## Data Availability

The data and model script are available in the Supplementary Files.

The water quality data from Barton Springs is available from the waterdata.usgs.gov repository for site 08155500: https://waterdata.usgs.gov/nwis/dv?cb_00010=on& cb_00060=on&cb_00095=on&cb_00300=on&cb_00400=on&cb_63680=on&format=rdb&site_no=08155500&referred_module=sw&period=&begin_date=2014-10-01&end_date=2018-12-31

## Supplemental Information

Supplemental information for this article can be found online at http://dx.doi.org/10.7717/peerj.11246#supplemental-information.

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
