# Peer review of "Subterranean movement inferred by temporary emigration in Barton Springs salamanders (Eurycea sosorum)"

_PeerJ, doi:10.7717/peerj.11246_

## Round 0.1 · original submission · Minor Revisions

Both reviewers found your study interesting and most results sound. I kindly ask you to address their comments in detail and submit a revised version.

Reviewer 1 ·

Basic reporting

In general I thought the description and presentation of the mark-recapture analysis was thorough and well done. The writing was clear throughout, and the use of references to support the authors’ thinking was good. I thought the paper was clear and informative.

Experimental design

The analysis of emigration obviously depends on the ability to reliably recognize individual salamanders. Because the recognition of individuals is so key to the interpretation of the data, I think it would be good to include some more details here. What was the similarity cutoff used to differentiate individuals? In your previous work, you described how image similarity declined with the time between captures of the same individual-is it possible or likely that the timing of these surveys and observed recaptures is sufficiently long that some individuals were not recognized as being recaptures, when they really were? Would that mean that the emigration rates you describe may be conservative? Is there anything else you can say about the image analysis to convince readers that the results are reliable?

The number of observers varied among surveys, and I am assuming the identity of the observers also varied. Both the number and identity of observers can have a big influence on counts, at least in theory. I think it would be good to mention in the methods if any of the observers were present for all (or most) surveys. Did you fit variables for number or identity of observers in any of the preliminary models? I think it would be worthwhile to try doing so-maybe it does not matter, but I can’t help but look at Figures 4 and 5 and wonder if any of the apparent trends through time are actually dependent on the number or identity of observers.

Paragraph beginning on line 192: here you explain why you don’t have directional hypotheses for the effect of discharge on emigration, but it is not clear that you have directional hypotheses for other variables either. Direction is implied by some of the explanations, but not all. Can you be explicit about expectations for the non-discharge variables?

Validity of the findings

Time was an important factor explaining both survival and emigration, but was not really discussed in detail in the discussion. For example: do you think there is any biological significance to the fact that emigration of individuals observed at the surface during the prior survey were the lowest during the forest surveys of 2016 & 2017 (orange circles in Figure 4C). Given the importance of time in the analysis, I think it makes sense to interpret the effect of time on the demographic parameters in the Discussion.

Reviewer 2 ·

Basic reporting

This paper was clear and well-written. The authors did a nice job of describing the need for the study in the Introduction/background and these sections were well referenced.

Experimental design

The research questions were well defined and the authors study design allowed them to assess these questions. Such studies of temporary emigration for facultative cave species are highly unusual and the quality of the authors data is high. The methods were described well, though I did get a little lost in the growth information - see specific comments below for description and recommended changes.

Validity of the findings

No comment - the paper meets standards.

Additional comments

This is a well-written paper on a difficult-to-study system. I commend the authors on their work and offer the following minor comments for them to consider to strengthen the current manuscript and things they might consider in future work. I have divided the section into editorial suggestions and ecological or methodological considerations.

Methodological/Ecological considerations

Line 170-71. You were not able to estimate superpopulation size because the temporary emigration was Markovian, right? If temporary emigration was random then you could estimate the superpopulation size (Kendall et al. 1997).

Lines 223-245. The use of the growth model confused me a bit. I thought the authors were going to use the growth model to predict size for each individual at each primary period so they could model survival and/or temporary emigration as a function of body size (continuous variable). Time-varying individual covariates are notoriously difficult to deal with in mark-recapture analysis because the size of individuals that are alive but not detection is unknown (as the authors state). External models, like von Bertalanffy growth models, are often used to predict the size of these individuals and those predictions are used to model parameters of interest (e.g., Hansen et al. 2015 Copeia). More sophisticated Bayesian approaches are available to jointly model the distribution of the covariate (e.g., size) and the capture histories (e.g., Bonner et al. 2010 Biometrics). However, the authors discretized the size predictions into large/adult (>24.2 mm) and small/juvenile (<24.2mm). If you were happy with a discretization of body size, then using a multi-state mark-recapture model seems much, much easier as it does not require size categories for uncaptured individuals and allows one to estimate and model the transition between size-states. Robust Design Multi-state mark-recapture models exist (e.g., in program MARK) and can be used to estimate movement to and from unobservable states (e.g., Kendall and Nichols 2002 J. Applied Statistics; Bailey et al. 2010 Ecology) and has been used in numerous amphibian studies (e.g., Church et al. 2007 Ecology). This approach would seem to be much easier and might allow for exploration of factors influencing transitions between juvenile and adult life stages. Importantly, I don’t think the authors approach is wrong, just more difficult than a multi-state approach if you were really only interested in two size classes.
Related to comment above. Lines 258 - here the authors state 333 salamanders were recaptured at least once between periods, but on line 270 they state that ‘growth was modeled from 901 measurements of 372 recaptured individuals’. Why is there seemingly two different values for the number of individuals recaptured over multiple periods?

Line 279. Shouldn’t this be ‘effect on the logit scale’? Survival and temporary emigration are probabilities and typically modeled on the logit, not the log scale.

Line 283-286. I recommend explaining these results a little better – in my experience, few readers understand the difference between gamma” and gamma’ (i.e., the difference between the two emigration parameters). I think this section would be better explained if you added some biological explanation. For example, you could say:
Temporary emigration estimates varied among size classes; the probability of moving from the pool to subterranean habitat was higher for larger individuals (range: …) than smaller individuals (range:…:Figure 4B). Similarly, the probability of remaining in the subterranean habitat was also higher for larger individuals (range: …) than smaller individuals (range:…; Figure 4C).
How were the average temporary emigration values calculated? Are these being reported from a model without time-variation or were they averaged over the point estimates? A measure of precision should be included with these estimates – this is easily obtained if the values are coming from a time-invariant model(s). The delta method would be necessary if you are averaging across point estimates.

Line 287. Clarify that this as the surface or pool abundance.

Lines 321-323. While time-varying models are more parsimonious that any of the models with environmental covariates they don’t give you much indication as to why there is variation in temporary emigration. Presumably, the environmental covariates were the most ecological plausible explanation, so even though those models are not as parsimonious as a full-time model, it would be nice to know if the direction of the relationships were consistent with your expectation. Your discussion suggests that there is temporal variation in movement, but your leading hypotheses (i.e., environmental covariate relationships) are never discussed or even presented…instead they are largely dismissed. Do you really want to do that?

Lines 331-333. If you used a multi-state approach in the future, you could model just juvenile survival as a function of flow. There is evidence for size-dependent ‘flushing’ in stream systems (e.g., Lowe et al. 2019 PNAS).

Lines 340-349. There is evidence of density-dependent movement between habitats or life-history states in other amphibians (e.g., Grayson et al. 2011 Ecology) and there are new models that might allow you to jointly model density-dependent survival (e.g., Kissel et al. 2020 Diversity)…again, thoughts for future research. It would be a great system to explore these possible processes.

Final thoughts: I think the authors do a good job of discussing possible mechanisms for their findings that smaller salamanders have lower emigration probabilities (e.g., flow velocities, differential effects of competition/predation, density-dependent effects), but it seems like the most compelling hypothesis given their findings is related to growth, reproduction, and avoiding predation. It would seem very logical that these salamanders, like many aquatic organisms, would show size-dependent growth, with young/small salamanders growing quickly in a short amount of time as illustrated by Fig 3. They would need to be in a resource-rich environment to do that, even though that environment might be risky. After they reach a given size threshold and sexual maturity, which seems to be reached around age 1 (Fig 3), then they migrate away from the surface to reproduce and stay in habitats with less predation. That is what the estimates seem to suggest – higher survival for larger individuals that are primarily in the subterranean habitat; i.e., these animals rarely return and if they do, they do not stay on the surface for very long (high gamma’ and gamma”). Smaller individuals are more likely to remain in the pool for more than one period and if they do emigrate, you really have no information, given the huge confidence intervals, in part because they are likely transition to the larger/adult size class. Such habitat switching is seen in the biphasic amphibians with aquatic larvae utilizing resource-rich aquatic habitats during high growth periods then metamorphosing to terrestrial life-phase with presumably lower/slower growth and lower predation risk.

Editorial and general suggestions:

Study Site section, Lines 174-175 and Fig 2. It would be nice to know who big the concrete pool is – the depth is given (line 125) but not the dimensions of the pool. How large are each formed outlet that convey the spring water? It would be nice add the black circle to figure 2B to better orient readers between the two figures in Fig 2. For example, in figure 2B is looks like the overland stream empties into the Barton Spring pool, but it must not – it flows into concrete pipes, right? Not apparent in the figure.

Line 183 -84. This sentence is a little confusing. You never fit covariate structures for the capture probabilities, right? In line 206 the authors state that period of high turbidity result in fine sediment depositions within the pool – does this influence capture probabilities? Consider replacing ‘optimal’ with ‘most-parsimonious’ in Lines 184 and 187..

Lines 189-191. Consider using these sentences as the start of the next paragraph because the next paragraph is really about that ‘final stage’ of model building/fitting. Perhaps remove these sentences and start the next paragraph with:
‘In the final model-building stage, we compared models where temporary emigration (gamma” and gamma’) varied over periods or as a function of environmental or individual covariates and models were survival probabilities varied with these same covariates. Specifically, we used five covariates…’
Lines 195-98. Consider wording to state: ‘Increases in flow may flush animals to the surface via drift, but flow may also prompt compensatory movement back into the spring outlets, influencing the rate of temporary migration. Permanent migration in response to discharge may also be reflected by apparent survival estimates.’

Lines 295. Consider replacing ‘run counter to’ with ‘contradict’

Lines 305. Add ‘this study’ to citations.

Line 324. Consider rewording – I don’t really understand this sentence.

Line 326. Consider replacing ‘determinants’ with ‘components’

---

## Round 0.2 · accepted · Accept

Congratulations, I am now pleased to accept your manuscript.

Reviewer 2 ·

Basic reporting

The authors addressed my previous comments very well.

Experimental design

The authors addressed my previous comments very well.

Validity of the findings

The authors addressed my previous comments very well.

Additional comments

The authors did a wonderful job addressing my previous comments!
I think they misspelled 'Grayson' in line 492, but that is the only issue that I have.
Super paper!